**Investigation**

# Functional classification of *GNAI1* disorder variants in *Caenorhabditis elegans* uncovers conserved and cell-specific mechanisms of dysfunction

Rehab Salama,[1] Eric Peet,[1] Thomas L. Morrione,[1] Sarah Durant,[1] Maxwell Seager (ID),[1] Madison Rennie (ID),[2]
Suzanne Scarlata,[2] Inna Nechipurenko (ID) [1,*]

[1]Department of Biology and Biotechnology, Worcester Polytechnic Institute, 100 Institute Road, Worcester, MA 01609, United States
[2]Department of Chemistry and Biochemistry, Worcester Polytechnic Institute, 100 Institute Road, Worcester, MA 01609, United States

*Corresponding author: Department of Biology and Biotechnology, Worcester Polytechnic Institute, 100 Institute Road, Worcester, MA 01609, United States.
Email: inechipurenko@wpi.edu

Heterotrimeric G proteins transduce signals from G protein–coupled receptors, which mediate key aspects of neuronal development and function. Mutations in the *GNAI1* gene, which encodes Gαi1, cause a disorder characterized by developmental delay, intellectual disability, hypotonia, and epilepsy. However, the mechanistic basis for this disorder remains unknown. Here, we show that *GNAI1* is required for ciliogenesis in human cells and use *Caenorhabditis elegans* as a whole-organism model to determine the functional impact of 7 *GNAI1*-disorder patient variants. Using CRISPR-Cas9 editing in combination with robust cellular (cilia morphology) and behavioral (chemotaxis) assays, we find that *T48I, K272R, A328P*, and *V334E* orthologous variants impact both cilia assembly and function in AWC neurons, *M88V* and *I321T* have no impact on either phenotype, and *D175V* exerts neuron-specific effects on cilia-dependent sensory behaviors. Finally, we validate in human ciliated cell lines that *D173V, K270R*, and *A326P GNAI1* variants disrupt ciliary localization of the encoded human Gαi1 proteins similarly to their corresponding orthologous substitutions in the *C. elegans* ODR-3 (*D175V, K272R*, and *A328P*). Overall, our findings determine the in vivo effects of orthologous *GNAI1* variants and contribute to the mechanistic understanding of *GNAI1*-disorder pathogenesis as well as neuron-specific roles of ODR-3 in sensory biology.

Keywords: *GNAI1*; ODR-3; Gα proteins; cilia; neurodevelopmental disorders

## Introduction

Heterotrimeric G proteins are key transducers of G protein–coupled receptor (GPCR) signaling, which plays key roles in neuronal development, communication, and behavior (Wettschureck and Offermanns 2005; Betke et al. 2012; Kurabayashi et al. 2013). In the canonical GPCR cascade, the activated receptor acts as a guanine-nucleotide exchange factor (GEF) to exchange guanosine diphosphate (GDP) on the Gα subunit of the Gαβγ trimer for guanosine triphosphate (GTP), thus stimulating dissociation of Gα-GTP from Gβγ and allowing activation of their respective downstream effectors (Gilman 1987; Pierce et al. 2002; Dror et al. 2015).

Many GPCRs, G proteins, and their downstream effectors localize to primary cilia—specialized cellular compartments that house molecular machinery of all major signaling pathways (Hilgendorf et al. 2016; Anvarian et al. 2019). Primary cilia mediate many aspects of neuronal biology in the developing and mature brain, including cell fate specification, proliferation, migration, axon guidance, dendrite morphogenesis, and neuronal excitability (Hasenpusch-Theil and Theil 2021; Suciu and Caspary 2021; Stoufflet and Caillé 2022; Jurisch-Yaksi et al. 2024). In humans, cilia dysfunction is associated with a spectrum of developmental disorders called ciliopathies (Hildebrandt et al. 2011; Reiter and Leroux 2017). Ciliopathy patients commonly exhibit neurological

symptoms that range in severity and include structural brain abnormalities, intellectual disability, motor deficits, and epilepsy (Lee and Gleeson 2011; Guemez-Gamboa et al. 2014; Valente et al. 2014), further highlighting the critical importance of primary cilia in the nervous system. In addition to classic ciliopathies, cilia defects are being increasingly reported in neurodevelopmental, neurodegenerative, and psychiatric disorders, suggesting that cilia dysfunction may be a shared feature of many neurological conditions (Karalis et al. 2022; Jurisch-Yaksi et al. 2024). However, the mechanisms by which cilia dysfunction contributes to neurological phenotypes observed in these disorders or those by which risk genes for these disorders impact cilia biology remain largely unknown.

Recent studies identified >15 de novo mutations in the *GNAI1* gene, which encodes the Gαi1 subunit of heterotrimeric G (αβγ) proteins, in individuals with a novel neurodevelopmental disorder (NDD) henceforth referred to as "*GNAI1* disorder" (Muir et al. 2021). Most affected individuals exhibited developmental delay, intellectual disability, hypotonia, and epilepsy—neurological symptoms that are also commonly associated with classic ciliopathies. Although the mechanistic basis for *GNAI1* disorder remains unknown, prior studies have described several neuronal functions for Gαi1. In the canonical GPCR cascade, Gαi1 is responsible for

decreasing cyclic adenosine monophosphate levels upon receptor activation by inhibiting adenylyl cyclases (Wettschureck and Offermanns 2005). Acute *GNAI1* knockdown in the mouse embryonic cortex disrupted multiple aspects of cortical development, including progenitor proliferation, neuronal migration, and dendritogenesis (Hamada et al. 2021). Additionally, genetic ablation of *GNAI1* has been reported to increase adenylyl cyclase activity and impair hippocampus-dependent long-term memory in mice (Pineda et al. 2004). While the cellular and molecular mechanisms that underlie neuronal functions of Gαi1 remain to be fully elucidated, these studies demonstrate that Gαi1 plays critical roles both in the developing and mature brain and underscore the importance of understanding the functional effects of *GNAI1* patient variants in relevant cellular and developmental contexts in vivo.

*Caenorhabditis elegans* constitutes a powerful whole-organism platform for rapidly evaluating phenotypic consequences of NDD-associated missense variants. Several recent studies successfully deployed *C. elegans* to functionally classify de novo variants in risk genes for developmental disorders that include *GNAO1* encephalopathy (Wang et al. 2022), autism-spectrum disorder (Wong et al. 2019), and ciliopathies (Lange et al. 2021, 2022). Here, we use *C. elegans* ODR-3, which belongs to the Gαi/o class of G proteins, as a model to evaluate the functional impacts of 7 NDD-associated *GNAI1* variants. Using *C. elegans* ODR-3 for this purpose offers several key advantages. First, all missense variants selected for analysis alter amino acids that are identical between Gαi1 and ODR-3 and map to the functional motifs conserved across Gα proteins. Second, *odr-3* loss-of-function (lf) and gain-of-function (gf) mutations result in robust and easily quantifiable cellular phenotypes—defects in cilia morphology in amphid wing A (AWA) and amphid wing C (AWC) olfactory neurons (Roayaie et al. 1998; Lans et al. 2004; Campagna et al. 2023). Finally, the *odr-3* function is required for a panel of chemosensory behaviors, including avoidance of high-osmolarity solutions detected by ASH and chemotaxis toward attractive odorants detected by AWA and AWC neurons (Bargmann et al. 1993; Roayaie et al. 1998). As a result, a complementary approach combining robust cellular and behavioral assays can be used to rapidly identify gene variants that are functionally consequential in vivo.

We find that 5 of the examined orthologous variants exhibited a range of defects in AWC cilia morphology and/or AWC-mediated attraction to benzaldehyde, while 2 variants appeared wild type in both assays. Interestingly, 1 examined variant had cell-specific effects on neuronal function—a marked deficit in ASH-mediated glycerol avoidance and unaltered AWC-mediated chemotaxis toward benzaldehyde. Finally, we confirm that the functional impact of 3 orthologous variants that severely compromised localization of the encoded ODR-3 proteins to AWC cilia is conserved in human cells.

# Materials and methods
## *C. elegans* strains and maintenance
All *C. elegans* strains were maintained at 15 °C or 20 °C on nematode growth medium (NGM) agar plates seeded with *Escherichia coli* OP50 (Brenner 1974). Standard genetic techniques were used to generate all strains. PCR and/or Sanger sequencing were used to confirm the genotypes of all mutant strains. Transgenic *C. elegans* carrying multicopy extrachromosomal arrays were generated by germline transformation. To generate transgenic strains, experimental plasmids were microinjected at 5 to 10 ng/μL together with either *unc-122Δp::gfp* or *unc-122Δp::dsred* co-injection markers injected at 30 and 40 ng/μL, respectively.

Wild-type and all mutant TagRFP-tagged *odr-3* constructs were injected at identical concentrations. Two or more independently generated lines were examined for all extrachromosomal transgenes, and the same extrachromosomal array was examined in wild-type and mutant backgrounds that were being directly compared. A list of *C. elegans* strains used in this study is provided in Supplementary Table 1.

## Chemotaxis behavioral assays
Population chemotaxis assays were performed on 10-cm round petri plates as previously described (Bargmann et al. 1993; Campagna et al. 2023). Briefly, 1 μL of benzaldehyde (Sigma) diluted in ethanol at 1:200, and 1 μL of ethanol (diluent control) were placed along with 1 μL of 1 M sodium azide anesthetic at opposite ends of the assay plate. Young adult hermaphrodites were washed off their growth plates with cholesterol-free S-basal, washed twice with S-basal, once with Milli-Q water, and transferred to assay plates. After 1 h, the chemotaxis index was calculated as follows:

$$\text{CI} = (\text{\#worms at attractant}-\text{\#worms at ethanol}) / \text{total worms that left origin}$$

All assays were performed on 3 or more days with at least 2 technical replicates per genotype per day. Wild-type and *odr-3(lf)* controls were included with experimental genotypes on all days of the assay.

## Osmotic avoidance assays
Osmotic avoidance assays were performed as previously described (Culotti and Russell 1978; Cornils et al. 2016). Briefly, 4 to 6 one-day-old hermaphrodites were transferred from a growth plate onto an NGM plate without food for 2 min to remove residual *E. coli* OP50. The same animals were subsequently placed inside a ring (~3/8th inch in diameter) of 8 M glycerol colored with bromophenol blue (USB). After 2 min, the number of animals that remained inside the glycerol ring was counted. All assays were repeated on 5 or more separate days with at least 3 technical replicates per genotype per day. Wild-type and *odr-3(lf)* controls were included with experimental genotypes on all days of the assay.

## CRISPR-Cas9 gene editing
Cas9 protein, crRNAs, and tracrRNA were purchased from Integrated DNA Technologies (IDT). All missense mutations and insertion of the split-wrmScarlet (wrmScarlet$_{11}$) tag were confirmed by Sanger sequencing.

### Generation of missense *odr-3* variants
Gene editing was carried out as described in Dokshin et al. (2018). Briefly, donor oligonucleotides carrying the desired missense mutations, silent mutations within the crRNA target sequence and/or PAM site to prevent repetitive cutting, and 36- to 40-bp homology arms were ordered from IDT (see Supplementary Fig. 4 for detailed sequence information; Supplementary File 1). The donor template (25 ng/μL), crRNA (56 ng/μL), tracrRNA (100 ng/μL), and Cas9 protein (250 ng/μL) were co-injected with *unc-122Δp::dsred* or *unc-122Δp::gfp* co-injection markers, and F1 progeny expressing the co-injection marker were isolated and genotyped for the presence of the edit. F2 individuals homozygous for the desired mutation were subsequently isolated from heterozygous F1 parents. All mutants generated via CRISRP-Cas9 editing were sequenced by Sanger sequencing and outcrossed at least 2 times to mitigate potential off-target effects.

### Generation of wrmScarlet₁₁ knock-in strain

To generate the split-wrmScarlet (*wrmScarlet₁₁*) allele *odr-3(nch013)*, a single-stranded donor oligonucleotide containing the *wrmScarlet₁₁* fragment (Goudeau et al. 2021), linkers, and homology arms was synthesized by IDT. *wrmScarlet₁₁* sequence was inserted between codons encoding Gly118 and Glu119 residues of ODR-3 (Campagna et al. 2023).

crRNA: 5′ GTACAGGAAAATGGAGAAGA 3′
donor oligonucleotide: 5′ AAGAAGCAGAAAAGGCAATAGTTA TGAAAGTACAGGAAAAcGGcGAgGAAGGAAGTGGAGGAGGA GGAAGTTACACCGTCGTCGAGCAATACGAGAA GTCCGTCGCCCGTCACTGCACCGGAGGAATGGATGAGTTATA CAAGAGTGGAGGAGGAGGAAGTGAAGCACTGACAGAAGAA GTTTCGAAAGCAATTCAATCG 3′

## Molecular biology

### odr-3 mutant constructs

Expression of *odr-3* transgenes in AWC and ASH was achieved by sub-cloning *odr-3* cDNA downstream from 0.7 kb of *ceh-36* (*ceh-36Δ*p) (Kim et al. 2010) and ~3 kb of *sra-6* (*sra-6*p) regulatory sequences, respectively. TagRFP tag was inserted between Gly118 and Glu119 residues of ODR-3 as previously described (Campagna et al. 2023). Patient mutations were introduced into plasmids carrying wild-type *tagrfp*-tagged *odr-3* cDNA downstream from cell-specific promoters by site-directed mutagenesis using QuikChange Lightning kit (Agilent Technologies). All constructs were verified by whole-plasmid sequencing (Plasmidsaurus).

### GNAI1 constructs

Wild-type coding sequence of human *GNAI1* (NM 002069) cloned into *pcDNA3.1+* (Invitrogen) mammalian expression vector was purchased from Bloomsburg University Foundation cDNA Resource Center (Catalog # GNAI100000). *eGFP* coding sequence with flanking DNA segments encoding GlyThr (5′ linker) and GlySer (3′ linker) was inserted between Leu91 and Lys92 of the *GNAI1* coding sequence in *pcDNA3.1+* vector using NEBuilder HiFi DNA assembly (NEB) to make *eGFP*-tagged *GNAI1* construct. Missense mutations corresponding to *D173V*, *K270R*, and *A326P* substitutions were introduced into the *pcDNA3.1+* plasmid containing *eGFP*-tagged *GNAI1* via site-directed mutagenesis using the QuikChange Lightning kit (Agilent Technologies). All constructs were verified by full-plasmid sequencing (Plasmidsaurus). A list of plasmids used in this work is provided in Supplementary Table 2.

### qPCR

Total RNA was extracted from siControl and si*GNAI1*-treated RPE-1 and HEK293T cells using the RNeasy kit (Qiagen) per manufacturer's instructions. RNA samples were reverse-transcribed (ZymoScript One-Step RT-qPCR Kit), and *GNAI1* expression (relative to *RPL11* control) was quantified by real-time PCR carried out on Applied Biosystems QuantStudio 6 Pro system using the $2^{-\Delta\Delta Ct}$ method. Primer pairs used in this study are as follows:

*RPL11*: 5′ GTTGGGGAGAGTGGAGACAG 3′ / 5′ TGCCAAAGGATCTGACAGTG 3′
*GNAI1*: 5′ CCCGAGAGTACCAGCTTAATG 3′ / 5′ CATCTTGTTGA GTCGGGATGTA 3′

## Cell culture and transfections

All cell lines were maintained at 37 °C with 5% $CO_2$ and tested monthly for mycoplasma using the mycoplasma PCR detection kit (ABM). Human telomerase-immortalized retinal pigment epithelial cells (hTERT RPE-1 [Nechipurenko et al. 2016]) were cultured in DMEM/F-12 (1:1) supplemented with 10% fetal bovine serum (FBS) and 1× antibiotic-antimycotic (Gibco). HEK293T cells (Abcam) were grown in DMEM high-glucose supplemented with Glutamax (Gibco), 10% FBS, and 1× antibiotic-antimycotic (Gibco).

### RNAi

For immunofluorescence analysis, RPE-1 and HEK293T cells were plated on 12-mm glass pretreated or poly-D-lysine-coated coverslips (Neuvitro), respectively, in 24-well plates at 30,000 cells per well in antibiotic-free complete-growth medium. For qPCR analysis, cells were plated without coverslips in 12-well plates at 60,000 cells per well. Synthetic small interfering RNA oligonucleotides (siRNAs) targeting non-overlapping *GNAI1* sequences or non-targeting control siRNA were transfected as previously described (Nechipurenko et al. 2016) using Lipofectamine RNAiMax (Invitrogen). The target sequences and sources of siRNAs used in this study are provided as follows:

si*GNAI1* #1 (s5872, Ambion): GAAUUGUUGAAACCCAUUU
si*GNAI1* #2 (J-010404-07-0002, Dharmacon): CAAAUUACAU CCCGACUCA
siControl (D-001810-01-05, Dharmacon): UGGUUUACAUGUC GACUAA

### Transfection of GNAI1 plasmids

HEK293T cells were plated on 12-mm glass poly-D-lysine-coated coverslips (Neuvitro) in 24-well plates at 30,000 cells per well in antibiotic-free complete-growth medium. Upon reaching 50% to 60% confluence, the plated cells were transfected with 500 ng of *pcDNA3.1+* vector containing *eGFP*-tagged *GNAI1^{WT}*, *GNAI1^{D173V}*, *GNAI1^{K270R}*, or *GNAI1^{A326P}* using Lipofectamine 3000 (Invitrogen) per manufacturer's instructions. After a 3-d incubation period, the transfection medium was replaced with serum-free medium to induce ciliation. Cells were serum-starved for 48 h prior to fixation and analysis.

## Immunostaining

RPE-1 and HEK293T cells were fixed in 4% paraformaldehyde for 12 min followed by permeabilization in 0.2% Triton X-100 (MP Biomedicals) for 10 min at room temperature (RT). Fixed cells were blocked with 5% bovine serum albumin in phosphate-buffered saline (PBS) with 0.2% Triton X-100 (PBS-T) for 1 h at RT or at 4 °C overnight, followed by incubation in primary antibodies for 1.5 h at RT or at 4 °C overnight. The following antibodies were used in immunofluorescence experiments: anti-γ-tubulin (clone 8D11, 1:500, Biorbyt, batch # T1578), anti-ARL13B (clone N295B/66, 1:10, Developmental Studies Hybridoma Bank), anti-acetylated α-tubulin (clone 6-11B-1, 1:500, Sigma, batch # T7451), anti-GFP (catalog # GFP-1010, 1:500, Aves Laboratories, batch # GFP917979), and anti-GM130/GOLGA2 (catalog # HPA021178, 1:500, Sigma, batch # A105115). Species-specific fluorescent secondary antibodies were obtained from Jackson ImmunoResearch Laboratories and used at a 1:500 dilution in PBS-T. DAPI (1:1,000, ThermoFisher) was used to stain DNA and applied together with secondary antibodies for 1.5 h at RT or at 4 °C overnight.

## Microscopy

### C. elegans

Synchronized 1-d-old adult hermaphrodites were anesthetized in 10 mM tetramisole hydrochloride (MP Biomedicals) and placed on a 10% agarose pad mounted on a microscope slide. Animals were

imaged on an upright THUNDER Imager 3D Tissue (Leica). Complete z-stacks of AWC and ASH sensory endings that included distal dendrite and cilia were acquired at 0.22-µm intervals with a K5 sCMOS camera (Leica) in Leica Application Suite X software using an HC Plan Apochromat 63× NA 1.4 to 0.60 oil immersion objective. To image 1-d-old split-wrmScarlet transgenic animals, single-plane snapshots were acquired on an inverted Nikon Ti-E microscope with Yokogawa CSU-X1 spinning disk confocal head using 60× NA 1.40 oil immersion objective.

### RPE-1 and HEK293T cells

Fixed and stained RPE-1 and HEK293T cells were mounted in ProLong Diamond antifade mountant (Invitrogen) and imaged on an inverted Nikon Ti-E microscope with Yokogawa CSU-X1 spinning disk confocal head using a 60× NA 1.40 oil immersion objective. Complete z-stacks were acquired at 0.25-µm intervals with an ORCA-Fusion BT Digital CMOS camera (Hamamatsu) in MetaMorph 7 software (molecular devices).

### Image analysis

Analysis of all fluorescence microscopy images was performed in Fiji/Image J (National Institute of Health, Bethesda, MD, USA). All data were quantified from images collected on at least 2 independent days. For experiments examining localization of TagRFP-tagged ODR-3 variants and eGFP-tagged Gαi1 variants, all genotypes/conditions being compared directly were imaged at identical settings.

### ODR-3::TagRFP fluorescence intensity (AWC and ASH)

TagRFP fluorescence intensity inside AWC cilia or periciliary membrane compartment (PCMC) in the distal dendrite was measured by outlining AWC cilia or PCMC using the freehand selection tool and measuring mean intensity inside the resultant ROIs. The relative fluorescence intensity was plotted as a ratio of mean TagRFP intensity inside the AWC cilium over mean TagRFP intensity inside the PCMC of the same neuron. TagRFP fluorescence intensity inside the ASH cilium was measured by drawing a line segment along the center of the ASH cilium (from base to tip) and measuring the mean intensity along the line. TagRFP intensity inside the PCMC of ASH was measured as described above for AWC, and the relative intensity was calculated and plotted as a ratio of intensity inside the ASH cilium over intensity in the PCMC of the same neuron.

### Cilia area (AWC)

The area of AWC cilia was measured as previously described (Campagna et al. 2023). Briefly, the cilia were outlined using the freehand selection tool, and the area enclosed by the ROI was measured from maximum-intensity projections that contained AWC cilia in their entirety.

### Gαi1::eGFP relative fluorescence intensity

Line segments were drawn from base to tip, along the center of the cilia, visualized using anti-ARL13B antibody, and mean intensity along the line was measured in the eGFP channel in transfected HEK293T cells. Another straight line was drawn across the Golgi, visualized with anti-GOLGA2 antibody, and the mean intensity along the line was measured in the eGFP channel. The relative Gαi1::eGFP intensity was calculated and plotted as a ratio of eGFP intensity inside the cilium over eGFP intensity in the Golgi of the same transfected cell.

### Statistical analysis

Prism 10 software (GraphPad, San Diego, CA, USA) was used to perform all statistical analyses and generate bar graphs and scatter plots. The D'Agostino–Pearson test was used to determine whether the data were normally distributed. Details on the numbers of analyzed animals and cells, statistical tests, and multiple comparison correction (when applicable) are presented in figure legends.

## Results

### GNAI1 is required for ciliogenesis in mammalian cells

Given the overlap in clinical features observed in classic ciliopathies and *GNAI1* disorder (NDD with hypotonia, impaired speech, and behavioral abnormalities; OMIM # 619854), we first wondered whether *GNAI1* may play a role in cilia assembly. To address this question, we knocked down *GNAI1* in human embryonic kidney (HEK293T) and immortalized retinal pigment epithelial (hTERT RPE-1) cells—2 commonly used cell culture models in cilia research. Roughly 80% of RPE-1 cells transfected with non-targeting control siRNA (siControl) possessed a primary cilium labeled with anti-ARL13B antibody; however, only 34% of cells transfected with si*GNAI1* (si*GNAI1* #1) that targets both *GNAI1* reference transcripts (NM 001256414.1 and NM 002069.5) were ciliated (Fig. 1a and c), suggesting that *GNAI1* is required for ciliogenesis in RPE-1 cells. Importantly, RPE-1 cells transfected with the second siRNA that targets a distinct non-overlapping region of *GNAI1* (si*GNAI1* #2) resulted in a similar knockdown efficiency and ciliation defect (Fig. 1c and e), suggesting that the observed cilia phenotype is likely *GNAI1*-dependent. To confirm that the reduction in cilia number observed upon *GNAI1* KD is not simply due to defective ciliary trafficking of ARL13B, we stained *GNAI1* KD RPE-1 cells with an antibody against acetylated tubulin, which labels the axoneme, and noted a comparable reduction in number of ciliated cells (Supplementary Fig. 1a and b; Supplementary File 1).

To test if the *GNAI1* function is broadly required for ciliogenesis in human cells, we next turned to HEK293T cells. In comparison to RPE-1 cells, HEK293T cells ciliate less robustly under equivalent growth conditions (Zhang et al. 2023; Mahajan et al. 2025). Roughly 40% of HEK293T cells treated with control siRNA were ciliated (Fig. 1d), and *GNAI1* KD in HEK293T cells caused a significant reduction in ciliogenesis, similarly to RPE-1 cells (Fig. 1b and d). However, the overall impact of *GNAI1* KD on ciliation of HEK293T cells appeared to be qualitatively milder relative to RPE-1 cells (compare Fig. 1c and d), despite similar *GNAI1* KD efficiency in both cell lines (Fig. 1e), indicating that *GNAI1* may differentially contribute to cilia assembly in different cell types.

### Selection of GNAI1-disorder variants for in vivo functional analysis

Sequence analysis determined that *C. elegans* ODR-3 exhibits 49% amino-acid sequence identity and 66% similarity to human Gαi1 (100% length, EMBOSS Needle; BLAST *e*-value $6 \times 10^{-127}$; Supplementary Fig. 2; Supplementary File 1). The majority of *GNAI1*-disorder missense variants impact residues in the protein domains conserved across all Gα proteins (Muir et al. 2021). Therefore, we reasoned that we could leverage the well-defined genetics of *odr-3* together with robust cellular and behavioral phenotypes displayed by *odr-3*(*lf*) and *odr-3*(*gf*) mutants to

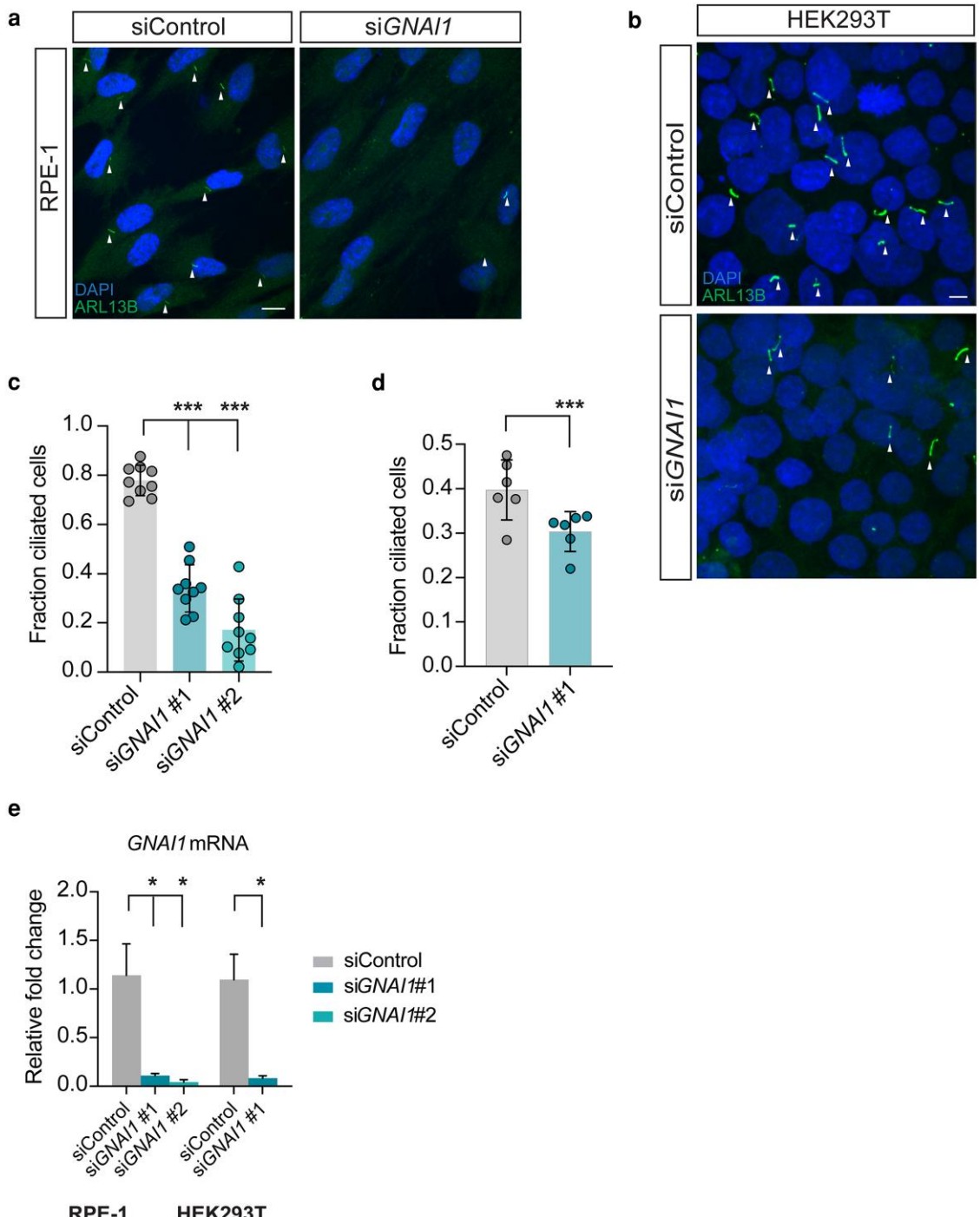

**Fig. 1.** *GNAI1* knockdown impairs ciliogenesis in human cells. a, b) Immunofluorescence images of fixed a) RPE-1 and b) HEK293T cells transfected with the indicated siRNAs. Arrowheads mark cilia labeled with anti-ARL13B antibody. Scale: 10 μm. c, d) Quantification of ciliation in c) RPE-1 and d) HEK293T cells transfected with the indicated siRNAs. Each data point represents 1 replicate. Total number of cells: siControl ($n = 858$ RPE-1, $n = 4,460$ HEK293T), si*GNAI1* #1 ($n = 518$ RPE-1, $n = 4,600$ HEK293T), si*GNAI1* #2 ($n = 400$). Means ± SD are indicated by shaded and vertical bars, respectively. ***Different from control at $P < 0.001$ (Fisher's exact test). e) Relative levels of *GNAI1* mRNA in RPE-1 and HEK293T cells treated with the indicated siRNAs. Summary data represent 4 replicates per condition per cell type. Error bars are SEM. *Different from control at $P < 0.05$ (Mann–Whitney test).

determine functional impacts of select *GNAI1* patient variants in vivo. To this end, we chose to focus on 7 missense *GNAI1* alleles that affect conserved, identical amino acids in ODR-3 (Fig. 2a and b; Supplementary Fig. 2; Supplementary File 1). Four mutations (*GNAI1*: T48I, D173V, K270R, and A326P) map to structurally conserved motifs called G boxes that mediate binding and hydrolysis of guanine nucleotides (Noel et al. 1993; Sprang 1997; Luo et al.

2022), 1 mutation (*GNAI1*: V332E) impacts a residue in the C-terminal alpha-helix, which participates in Gα interactions with GPCRs (Oldham et al. 2006; Masuho et al. 2023) and cytoplasmic GEF RIC-8 (Kant et al. 2016; Zeng et al. 2019; Seven et al. 2020), and 2 mutations (*GNAI1*: M88V and I319T) are in the conserved residues outside of known functional domains (Fig. 2a and b).

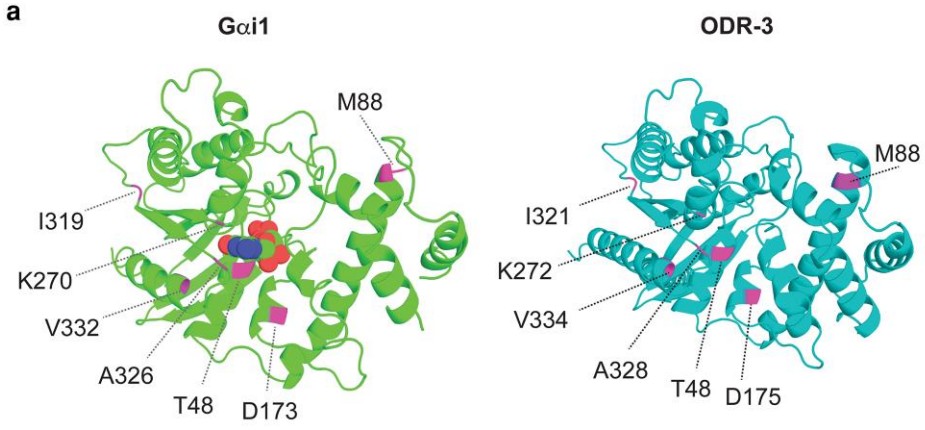

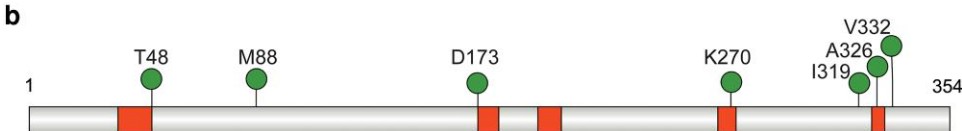

**Fig. 2.** Distribution of patient variants selected for analysis. a) *Left*: 3D structure of Gαi1 bound to GDP (PDB accession: 2G83) (Johnston et al. 2006). Carbon, oxygen, nitrogen, and phosphorous atoms of GDP are shown as green, red, blue, and orange spheres, respectively. *Right*: AlphaFold-predicted structure of *C. elegans* ODR-3 (Varadi et al. 2022, 2024). Positions of missense patient variants that were selected for analysis are shown in magenta in both protein models. The protein structures for Gαi1 and ODR-3 were generated in the PyMOL Molecular Graphics System, Version 3.1.4 (Schrodinger, LLC). b) Schematic of human Gαi1 showing positions of *GNAI1*-disorder-associated variants (pins). Guanine nucleotide-binding motifs (G1 to G5 boxes) are shown in red. The Gαi1 protein diagram was generated using IBS 2.0 (Xie et al. 2022).

## GNAI1-disorder-associated orthologous variants differentially impact ODR-3 ciliary localization

Previous studies used overexpressed ODR-3 transgenes and immunofluorescence against endogenous ODR-3 to demonstrate that wild-type ODR-3 (ODR-3$^{WT}$) is normally enriched in the cilia of AWC sensory neurons (Roayaie et al. 1998). First, we wanted to determine whether any of the selected variants affect the subcellular localization of ODR-3. To address this question, we first tagged the *odr-3* locus with split-wrmScarlet reporter (wrmScarlet$_{11}$) (Goudeau et al. 2021) using CRISPR-Cas9 genome editing (Supplementary Fig. 3a; Supplementary File 1). To visualize endogenous ODR-3 in AWC cilia, we reconstituted fluorescence via expression of the wrmScarlet$_{1-10}$ fragment in AWC neurons under the *ceh-36Δ* promoter (Kim et al. 2010) (Supplementary Fig. 3a; Supplementary File 1). Consistent with published studies, ODR-3$^{WT}$::split-wrmScarlet was enriched inside AWC cilia (*n* = 25 animals, Supplementary Fig. 3b; Supplementary File 1). However, the reconstituted wrmScarlet was photobleached nearly instantaneously upon sample illumination with the laser light, making it impossible to acquire multiple z-slices necessary for in-depth quantitative analysis of ODR-3 subcellular localization.

To bypass the photobleaching issue, we next expressed TagRFP-tagged wild-type or mutant ODR-3 variants from transgenes in AWC neurons of wild-type animals and examined cilia localization of the encoded proteins. Similarly to the endogenous ODR-3, ODR-3$^{WT}$::TagRFP expressed from a multicopy transgene was enriched in AWC cilia (Campagna et al. 2023; Fig. 3b and c). We noted no significant changes in subcellular localization of TagRFP-tagged ODR-3$^{I321T}$ or ODR-3$^{V334E}$ variants (Fig. 3b and c). Interestingly, we observed 2 distinct patterns of mislocalization amongst the remaining ODR-3 protein variants. Like ODR-3$^{WT}$,

ODR-3$^{T48I}$ and ODR-3$^{M88V}$ were present inside the AWC cilium. However, unlike the WT protein, these variants were also detected in an ectopic pool in the PCMC of the distal dendrite (Fig. 3a to c). Finally, ODR-3$^{D175V}$, ODR-3$^{K272R}$, and ODR-3$^{A328P}$ exhibited the most striking defects in subcellular localization—these mutant proteins were largely excluded from the AWC cilium and instead accumulated in the PCMC (Fig. 3b and c).

To rule out the possibility that dendritic mislocalization observed with the mutant variants was an artifact of transgene overexpression, we introduced the *A328P* mutation into the *odr-3::split-wrmScarlet* background. The endogenously tagged ODR-3$^{A328P}$::split-wrmScarlet exhibited cilia localization defects that were qualitatively similar to those observed in transgenic animals overexpressing ODR-3$^{A328P}$::TagRFP in AWC neurons (Supplementary Fig. 3b; Supplementary File 1). Specifically, ODR-3$^{A328P}$::split-wrmScarlet was present throughout the AWC dendrite in all examined animals (*n* = 30) and prominently accumulated at the cilia base in 28 out of 30 analyzed animals (Supplementary Fig. 3b; Supplementary File 1). Collectively, these data indicate that *T48I*, *M88V*, *D175V*, *K272R*, and *A328P* variants affect ciliary trafficking of the encoded ODR-3 proteins, with the latter 3 mutations having the most profound impact on ODR-3 ciliary localization.

## The transition zone contributes to ciliary exclusion of the ODR-3$^{A328P}$ protein

Protein trafficking in and out of primary cilia is tightly controlled by the transition zone (TZ)—a selective diffusion barrier at the cilia base (Garcia-Gonzalo and Reiter 2017). One possible explanation for the exclusion of ODR-3$^{A328P}$, ODR-3$^{K272R}$, and ODR-3$^{D175V}$ from the AWC cilium with a concomitant accumulation in the distal dendrite is that these mutations impede ODR-3

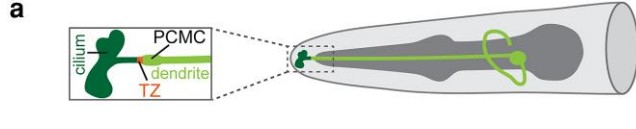

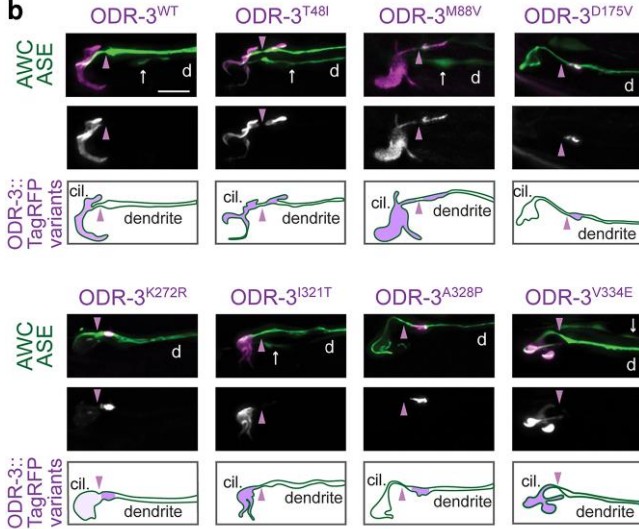

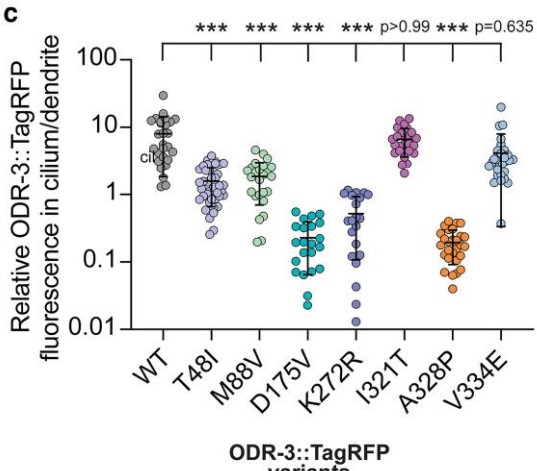

**Fig. 3.** Orthologous patient mutations in the *odr-3* gene differentially affect subcellular localization of the encoded proteins. a) Diagram of the AWC olfactory neuron in the worm nose (only 1 AWC of a pair is shown for simplicity). The region inside of the dashed box is magnified in a panel on the left. TZ, transition zone; PCMC, periciliary membrane compartment. b) Representative images and cartoon summaries (bottom rows) showing localization of the indicated TagRFP-tagged ODR-3 variants in AWC neurons. AWC neurons were visualized via expression of *ceh-36*p::GFP, which also labels ASE neurons (arrows). Arrowheads mark cilia base/TZ. d, dendrite. Anterior is at the left. Scale: 5 μm. c) Quantification of relative TagRFP fluorescence inside the AWC cilium vs distal dendrite for the indicated TagRFP-tagged ODR-3 variants. Total number of AWC neurons analyzed for each variant: WT ($n = 27$), T48I ($n = 42$), M88V ($n = 26$), D175V ($n = 22$), K272R ($n = 21$), I321T ($n = 26$), A328P ($n = 28$), and V334E ($n = 29$). Means ± SD are indicated by horizontal and vertical bars, respectively. ***Different from WT at $P < 0.001$ (Kruskal–Wallis with Dunn's multiple comparisons test).

transport into the cilium across the TZ. If this hypothesis were true, we would expect to see an increase in intraciliary levels of ODR-3^A328P protein in animals with disrupted TZ architecture

compared with wild type. To test this hypothesis, we examined the localization of ODR-3^A328P in *mks-5(tm3100)* mutants. MKS-5 is a core component of the TZ, and *mks-5* mutants exhibit severely compromised TZ integrity (Jensen et al. 2015). In line with our hypothesis, we noted an increase in ODR-3^A328P::TagRFP levels inside the cilium relative to distal dendrite in *mks-5(tm3100)* mutants compared with ODR-3^A328P::TagRFP levels in the wild-type background (Fig. 4a to c). Notably, when we examined localization of ODR-3^WT::TagRFP in *mks-5(tm3100)* mutants, we noticed that in addition to cilia, ODR-3^WT::TagRFP was now present in the distal dendrite (Fig. 4a to c), consistent with the previously published role for the TZ in preventing leakage of membrane-associated ciliary proteins into the dendrite (Cevik et al. 2013). Collectively, our results suggest that the TZ plays an important part in regulating ODR-3 ciliary entry and retention, and that the *A328P* variant likely impacts ODR-3 trafficking across the TZ.

### *odr-3* Mutations orthologous to patient *GNAI1* variants differentially affect AWC cilia morphology

To gain insight into the functional effects of the *GNAI1*-disorder-associated variants in vivo, we used CRISPR-Cas9 editing to engineer worms homozygous for orthologous variants in the *odr-3* gene (Supplementary Figs 4 and 5; Supplementary File 1). To visualize AWC cilia morphology, which is dependent on intact *odr-3* function, we then crossed an AWC-specific *gfp* reporter into the CRISPR-Cas9-engineered *odr-3* mutant strains. Consistent with published work, *odr-3(lf)* animals exhibited a marked reduction in the AWC cilia area compared with wild type (Fig. 5a and b). Cilia appeared normal in *M88V* and *I321T* homozygous animals (Fig. 5a and b). On the other hand, *T48I*, *D175V*, *A328P*, and *V334E* mutants exhibited cilia defects that ranged from a mild albeit significant reduction in cilia size in *T48I* and *V334E* mutants to markedly smaller cilia in *D175V* and *A328P* mutants (Fig. 5a and b). Notably, the cilia phenotype in *A328P* animals was qualitatively and quantitatively indistinguishable from that of *odr-3(lf)* mutants, suggesting that *A328P* allele may be a functional null. Re-expression of wild-type *odr-3* in AWC neurons of *A328P* mutant animals partially but significantly rescued AWC cilia morphology (Fig. 5a and b), indicating that cilia defects in *A328P* mutants are indeed due to the loss of *odr-3* function. Despite multiple attempts, we were unable to engineer a *K272R* variant in the *odr-3* gene; so we took an alternative approach to determine whether this mutation impacts *odr-3* function. Specifically, we overexpressed ODR-3^K272R cDNA in AWC neurons of *odr-3(lf)* animals and quantified AWC cilia size in the transgenic animals. We found that these transgenic worms were indistinguishable from *odr-3(lf)* mutants (Fig. 5a and b), suggesting that the *K272R* mutation strongly impairs ODR-3 activity.

*K272R* and *A328P*—2 mutations that result in the most severe cilia defects—alter conserved amino acids that come into direct contact with GDP/GTP. Lysine 272 participates in guanosine recognition (Luo et al. 2022), while alanine 328 is found in the TCAT motif, which is conserved throughout the G protein family, and forms a hydrogen bond with the oxygen in guanine (Luo et al. 2022). Therefore, cilia morphology defects observed in *K272R* and *A328P* mutants likely stem from aberrant interaction of ODR-3 with guanosine nucleotides.

### The *A328P* variant does not abolish ODR-3 interaction with RIC-8 or UNC-119

We previously showed that ODR-3 function is regulated by the cytoplasmic GEF and chaperone RIC-8 (Campagna et al. 2023).

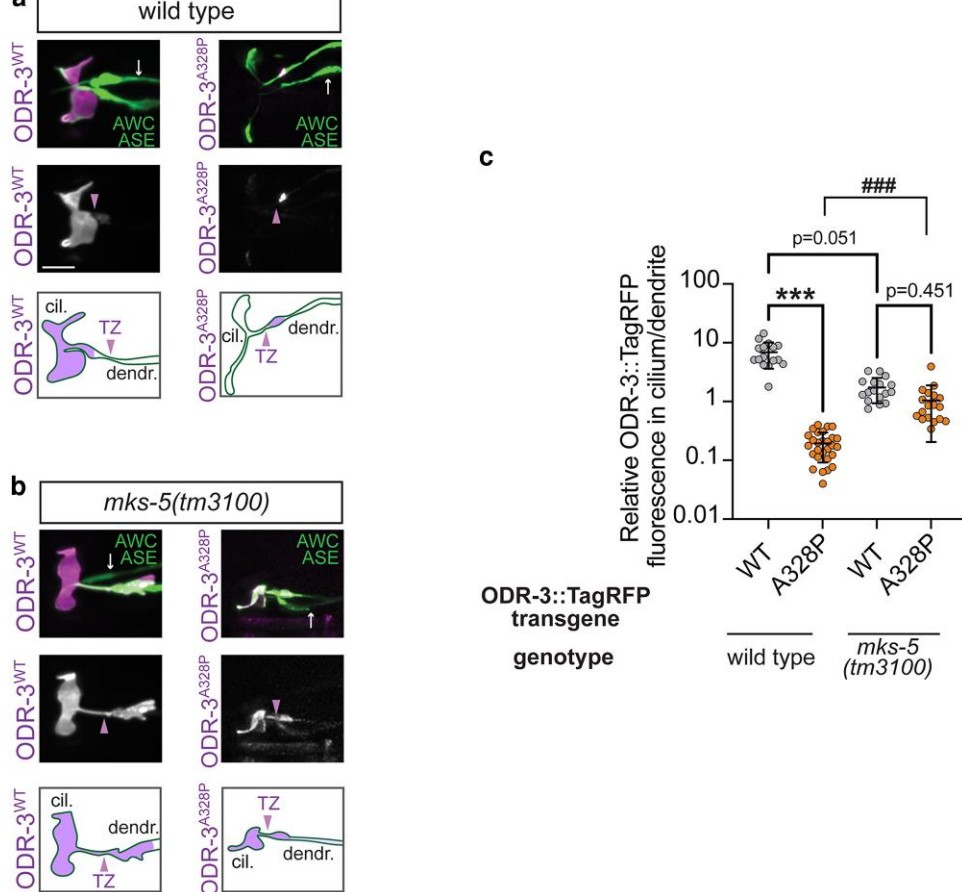

**Fig. 4.** Intraciliary localization of A328P ODR-3 variant is increased in *mks-5* mutants. a, b) Representative images (top and middle) and cartoons (bottom) showing localization of TagRFP-tagged ODR-3^WT and ODR-3^A328P proteins in AWC cilia of a) wild-type animals and b) *mks-5(tm3100)* mutants. AWC neurons were visualized via expression of *ceh-36p::GFP*, which also labels ASE neurons (arrows). Arrowheads mark cilia base/TZ. cil., cilium; dendr., dendrite. Anterior is at the left. Scale: 5 µm. c) Quantification of relative TagRFP fluorescence inside the AWC cilium vs distal dendrite for the indicated TagRFP-tagged ODR-3 variants in wild-type and *mks-5(tm3100)* mutant animals. Total number of AWC neurons analyzed for each variant: wild type ($n = 16$ ODR-3^WT, $n = 28$ ODR-3^A328P), *mks-5(tm3100)* ($n = 17$ ODR-3^WT, $n = 18$ ODR-3^A328P). Data for ODR-3^WT localization in the wild-type background are repeated from Fig. 3c. Means ± SD are indicated by horizontal and vertical bars, respectively. *** and ###Different between bracketed groups at $P < 0.001$ (Kruskal–Wallis with Dunn's multiple comparisons test).

Like *odr-3(lf)* and A328P mutants, *ric-8(lf)* mutants exhibit a marked reduction in AWC cilia size. Since the *A328P* variant alters a residue just proximal to the Gα α5 helix, which participates in Gα—RIC-8 binding (Kant et al. 2016; Zeng et al. 2019), and most closely resembles *odr-3(lf)*, we chose to focus on this variant to determine whether it may impact cilia morphology by interfering with the ODR-3—RIC-8 interaction. To address this question, we first examined in vivo association between ODR-3^A328P and RIC-8 via bimolecular fluorescence complementation (BiFC) (Shyu et al. 2008) (see Supplementary File 1 for Materials and Methods). We found that while both ODR-3^WT and ODR-3^A328P variants tagged with the N-terminal Venus fragment (VN) associated with RIC-8 tagged with the C-terminal half of Venus (VC) (Supplementary Fig. 6a and b; Supplementary File 1), the percentage of animals exhibiting BiFC was lower when RIC-8 was co-expressed with ODR-3^A328P compared with ODR-3^WT (Supplementary Fig. 6b; Supplementary File 1). Notably, fluorescent signal was frequently detected at the cilia base rather than inside the cilium in animals co-expressing RIC-8 and ODR-3^A328P(compare middle and bottom panels in Supplementary Fig. 6a, Supplementary File 1), consistent with accumulation of the endogenously tagged ODR-3^A328P::split-wrmScarlet at cilia base (see Supplementary Fig. 3; Supplementary File 1).

Since BiFC is irreversible and may increase the likelihood of false positives over time (Kerppola 2008; Xing et al. 2016), we next turned to FRET-FLIM (Algar et al. 2019; (see Supplementary File 1 for Materials and Methods) to confirm that ODR-3^A328P does indeed bind RIC-8 in vivo. We expressed GFP-fused RIC-8 alone or together with TagRFP-tagged ODR-3^WT or ODR-3^A328P constructs in AWC neurons under the *ceh-36Δ* promoter. As compared with RIC-8::GFP alone, the lifetime of GFP decreased in animals co-expressing RIC-8::GFP with ODR-3^WT confirming interaction between the 2 proteins (Supplementary Fig. 6c and d; Supplementary File 1). Comparable reduction in GFP lifetime was also observed when we co-expressed RIC-8::GFP with ODR-3^A328P (Supplementary Fig. 6c and d; Supplementary File 1), suggesting that *A328P* substitution does not abolish the association of the mutant ODR-3 with RIC-8, consistent with our BiFC data.

The GTP/GDP exchange cycle was also proposed to regulate the interaction between Gα proteins and an evolutionarily conserved trafficking adaptor UNC-119, which plays a key role in transporting Gα proteins to cilia (Zhang et al. 2011). Studies in mice and *C. elegans* demonstrated that *unc-119* deletion causes Gα protein mislocalization in sensory neurons of both species (Zhang et al. 2011). Furthermore, knocking down or mutating *unc-119* in

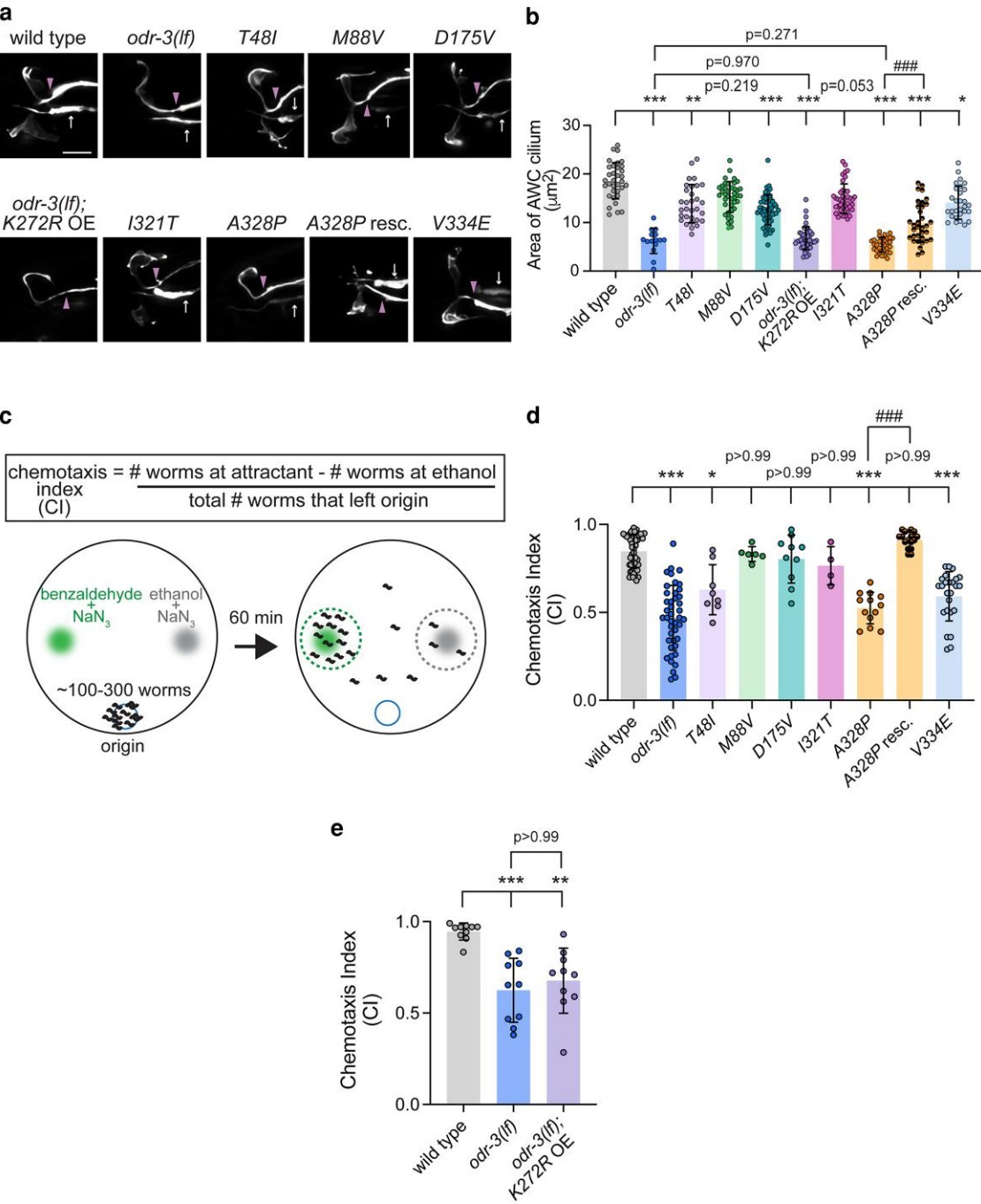

**Fig. 5.** Orthologous patient mutations in the *odr-3* gene differentially alter AWC cilia morphology and function. a) Representative images of AWC cilia in animals homozygous for the indicated *odr-3* mutations and *odr-3*(*lf*) animals overexpressing the *odr-3*$^{K272R}$ transgene. AWC neurons were visualized via expression of *ceh-36p::GFP*, which also labels ASE neurons (arrows). Arrowheads mark cilia base/TZ. Anterior is at the left. Scale: 5 μm. b) Quantification of AWC cilia size in the indicated genotypes. Total number of cilia: WT (*n* = 34), *odr-3*(*lf*) (*n* = 15), T48I (*n* = 31), M88V (*n* = 40), D175V (*n* = 59), K272R (*n* = 45), I321T (*n* = 41), A328P (*n* = 32), A328P rescue (*n* = 32), and V334E (*n* = 29). Means ± SD are indicated by shaded and vertical bars, respectively. *, **, and ***Different from WT at *P* < 0.05, 0.01, and 0.001, respectively (Kruskal–Wallis with Dunn's multiple comparisons test). ###Different between bracketed genotypes at *P* < 0.001 (unpaired *t* test with Welch's correction). c) Schematic of chemotaxis assay. d) Chemotaxis responses of the indicated homozygous variants. Individual data points represent chemotaxis index from 1 assay. Total number of assays: WT (*n* = 50), *odr-3*(*lf*) (*n* = 43), T48I (*n* = 8), M88V (*n* = 6), D175V (*n* = 10), I321T (*n* = 4), A328P (*n* = 13), A328P rescue (*n* = 16), and V334E (*n* = 25). Means ± SD are indicated by shaded and vertical bars, respectively. * and ***Different from wild type at *P* < 0.05 and 0.001, respectively (Kruskal–Wallis with Dunn's multiple comparisons test). ###Different between bracketed genotypes at *P* < 0.001 (unpaired *t* test with Welch's correction). e) Chemotaxis responses of wild-type, *odr-3*(*lf*), and *odr-3*(*lf*) mutants overexpressing *odr-3*$^{K272R}$ transgene in AWC neurons. Individual data points represent chemotaxis index from 1 assay. Total number of assays: WT (*n* = 10), *odr-3*(*lf*) (*n* = 10), and *odr-3*(*lf*); K272 OE (*n* = 10). Means ± SD are indicated by shaded and vertical bars, respectively. ** and ***Different from wild type at *P* < 0.01 and 0.001, respectively (Kruskal–Wallis with Dunn's multiple comparisons test).

zebrafish and *C. elegans*, respectively, produced phenotypes consistent with aberrant cilia function (Wright et al. 2011). Since A328 directly binds guanosine nucleotides, and mutations in

this residue interfere with Gα—GDP association (Posner et al. 1998), we wondered next whether the *A328P* variant may disrupt ODR-3 interaction with UNC-119. The loss of UNC-119—ODR-3

**Table 1.** Summary of AWC cellular and behavioral phenotypes caused by orthologous mutations in the *odr-3* gene.

| ODR-3 variant | T48I | M88V | D175V | K272R | I321T | A328P | V334E |
|---|---|---|---|---|---|---|---|
| Functional domain | G box | Unknown | G box | G box | Unknown | G box | Putative GPCR-binding |
| Encoded variant localization | Cilia + PCMC | Cilia + PCMC | Low inside cilia/pool at base | Wild-type | | Low inside cilia/pool at base | Wild-type |
| AWC cilia area | Reduced | Wild-type | Reduced | | Wild-type | Reduced | Reduced |
| Chemotaxis behavior | Impaired | Wild-type | Wild-type | Impaired | Wild-type | Impaired | Impaired |

association would also explain the accumulation of ODR-3$^{A328P}$ protein in the dendrite as well as severe cilia defects observed in *A328P* mutants. We expressed UNC-119 tagged with the C-terminal fragment of Venus (VC::UNC-119) together with either ODR-3$^{WT}$ or ODR-3$^{A328P}$ tagged with the N-terminal Venus fragment (ODR-3$^{WT}$::VN and ODR-3$^{A328P}$::VN, respectively) in AWC neurons of wild-type animals. We observed BiFC in 28% of animals co-expressing VC::UNC-119 and free N-terminal Venus fragment (negative control) (Supplementary Fig. 7a and b; Supplementary File 1). Co-expression of VC::UNC-119 with ODR-3$^{WT}$::VN or ODR-3$^{A328P}$::VN reconstituted fluorescent signal in AWC neurons of a significantly greater percent of transgenic animals (81% and 51%, respectively) (Supplementary Fig. 7b; Supplementary File 1), indicating that ODR-3$^{A328P}$ retains the ability to interact with UNC-119. Similarly to the BiFC assay with RIC-8 and ODR-3$^{A328P}$, the BiFC signal in animals co-expressing VC::UNC-119 with ODR-3$^{A328P}$::VN appeared to concentrate at the cilia base rather than being distributed throughout the cilium fan (Supplementary Fig. 7a; Supplementary File 1), and the percentage of animals exhibiting BiFC was lower when UNC-119 was co-expressed with ODR-3$^{A328P}$ compared to ODR-3$^{WT}$ (Supplementary Fig. 7b; Supplementary File 1). These results are consistent with the hypothesis that ciliary transport of the ODR-3-UNC-119 complex may be compromised by the *A328P* mutation. Additionally, a reduced percentage of animals that exhibit BiFC when ODR-3$^{A328P}$ is co-expressed with either RIC-8 or UNC-119 may be indicative of a weaker association between the mutant ODR-3 and these proteins when compared with wild-type ODR-3.

## The *T48I*, *K272R*, and *A328P* variants in the guanine-binding domains and *V334E* variant in the putative receptor-binding motif impair AWC-mediated chemotaxis

We next examined the impact of orthologous variants on sensory responses mediated by AWC. Ciliated endings of AWC neurons are enveloped in the glial processes (Perkins et al. 1986; Doroquez et al. 2014) and house signaling machinery that detects a panel of volatile attractants, including benzaldehyde (Bargmann et al. 1993; Ferkey et al. 2021). However, prior studies reported that these primary behavioral responses to volatile odorants are preserved in several mutants with severely compromised cilia morphology (Campagna et al. 2023; Philbrook et al. 2024), thereby uncoupling cilia morphology from cilia function in mediating odorant sensing by AWC. To investigate the impact of orthologous patient variants on ODR-3-dependent attraction toward benzaldehyde, we carried out population chemotaxis assays (Fig. 5c). As expected, wild-type animals exhibited robust attraction to benzaldehyde (1:200), while *odr-3(lf)* animals were deficient in this response (Roayaie et al. 1998; Fig. 5d). *M88V* and *I321T* mutants exhibited normal chemotaxis, suggesting that these variants do not impair ODR-3 function in either AWC ciliogenesis or chemotaxis toward benzaldehyde. On the other hand, *D175V* mutants exhibited normal behavioral response (Fig. 5d), despite having

morphologically defective AWC cilia (see Fig. 5a and b). Normal chemotaxis of *D175V* mutants was particularly intriguing given the dramatic impact of this variant on ciliary localization of the encoded ODR-3 protein (see Fig. 3b), and suggests that ODR-3 enrichment inside the AWC cilium may be important for its role in mediating ciliogenesis but not benzaldehyde attraction.

The variants mapping to the nucleotide-binding domain all resulted in impaired chemotaxis toward benzaldehyde. Specifically, *A328P* and *T48I* homozygous mutants as well as transgenic animals expressing *odr-3$^{K272R}$* cDNA in *odr-3(lf)* AWC neurons exhibited significantly lower chemotaxis indices compared with wild type (Fig. 5d and e), consistent with the hypothesis that these variants disrupt the overall signaling function of ODR-3. Of note, the *V334E* variant, which alters an amino acid in the putative GPCR-interaction motif, also negatively impacted ODR-3-mediated behavior (Fig. 5d), suggesting that this missense mutation may disrupt ODR-3 association with GPCRs. The summary of all cellular and behavioral phenotypes exhibited by the orthologous patient variants is provided in Table 1.

## The *D175V* variant disrupts ODR-3 ciliary localization and sensory function of ASH neurons

In addition to AWC, *odr-3* is expressed in several sensory neurons, including the nociceptor ASH that detects a range of aversive chemicals (Roayaie et al. 1998). Similarly to AWC, *odr-3* mediates chemosensory transduction in ASH neurons (Bargmann et al. 1993; Roayaie et al. 1998; Yoshida et al. 2012). However, *odr-3* function appears to be dispensable for ASH cilium assembly (Roayaie et al. 1998; Supplementary Fig. 8; Supplementary File 1), pointing to cell-specific roles for this gene in cilia biology.

Unlike AWC, ASH neurons possess simple rod-like cilia that extend through a glial channel and are exposed to the external environment (Perkins et al. 1986; Doroquez et al. 2014; Fig. 6a). A recent study demonstrated that a minimum ASH cilium length is required for ASH-mediated sensory responses to aqueous chemicals, including behavioral avoidance of high-osmolarity solutions (Philbrook et al. 2024). In this context, the minimum length is likely necessary to allow for direct contact between aqueous cues and cilia-localized signaling machinery. Since the *D175V* variant precludes the encoded ODR-3 protein from entering the cilium without abolishing Gα protein signaling function (ie *D175V* mutants can still respond to benzaldehyde, see Fig. 5d), we chose this variant to determine whether ODR-3 localization to the cilium is required for its function in ASH. We reasoned that if ODR-3 must be present inside the ASH cilium to mediate sensory transduction, then the *D175V* mutation may exert distinct effects on the function of ASH and AWC neurons.

First, we confirmed that TagRFP-tagged ODR-3$^{WT}$ expressed under the *sra-6* regulatory sequences localized to ASH cilia in wild-type animals (Fig. 6b). Like in AWC, ODR-3$^{D175V}$ expressed in ASH neurons of wild-type *C. elegans* was largely excluded from cilia and instead accumulated in the distal dendrite

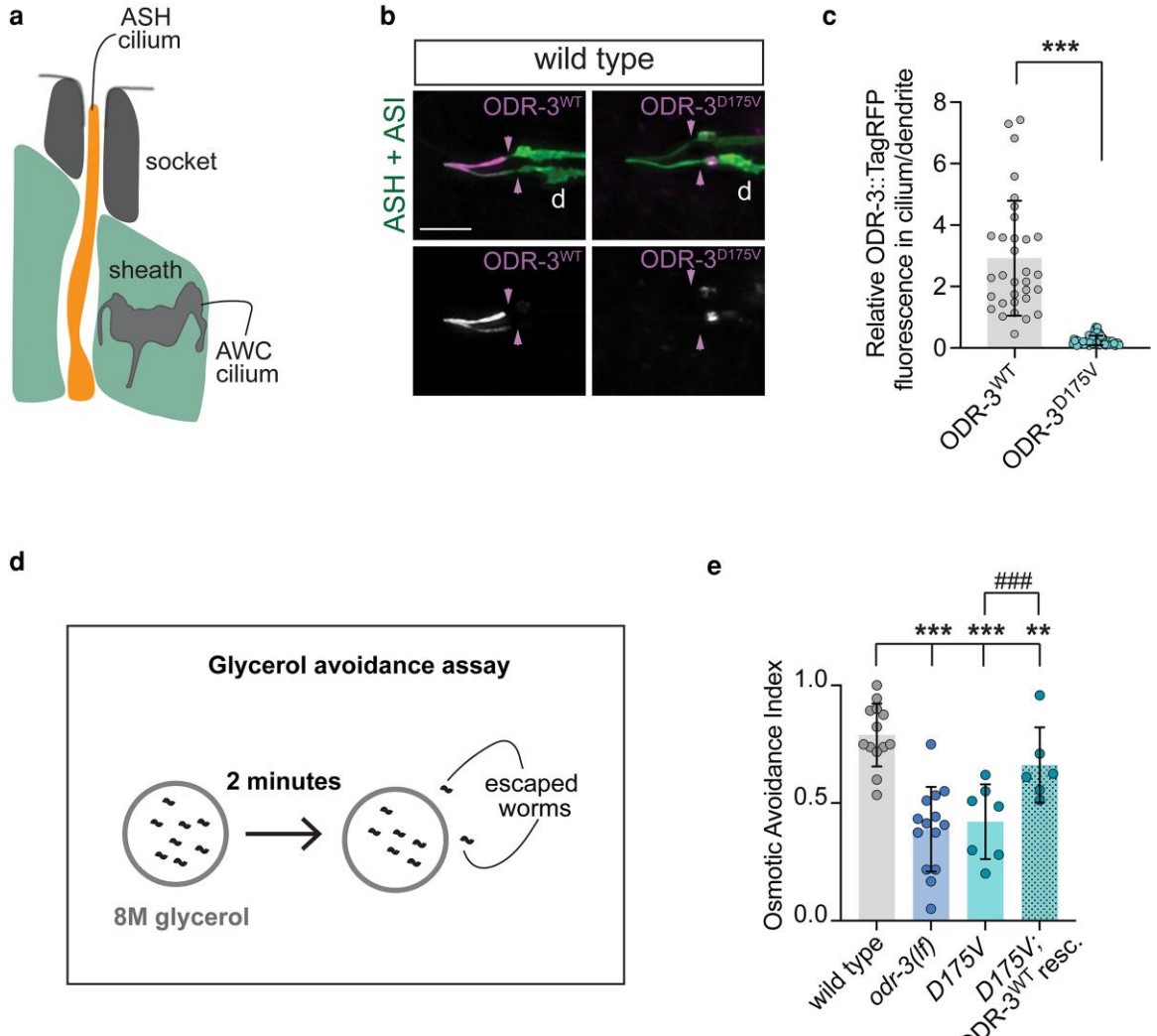

**Fig. 6.** The *D175V* variant disrupts ODR-3 localization to the ASH cilia and impairs ASH-mediated behavior. a) Schematic of the amphid channel. Only AWC and ASH cilia are shown for simplicity. Socket and sheath are glial cells that form the channel. Anterior is up. b) Representative images showing localization of TagRFP-tagged WT and D175V variants of ODR-3 in ASH and ASI neurons. ASH and ASI neurons were visualized via expression of *sra-6p::* GFP. Arrowheads mark cilia base/TZ. Anterior is at the left. Scale: 5 μm. c) Quantification of relative TagRFP fluorescence inside the ASH cilium vs distal dendrite for the indicated TagRFP-tagged ODR-3 variants. Total number of ASH neurons analyzed for each variant: WT ($n = 32$), D175V ($n = 65$). ***Different from WT at $P < 0.001$ (Mann–Whitney test). d) Diagram of glycerol avoidance assay. e) Fraction of animals that remain inside of an 8M-glycerol ring after 2 min (osmotic avoidance index). Genotypes of assayed animals are indicated. Each data point represents a mean osmotic index per genotype per day. Three to 8 technical replicates were assayed per genotype per day. Total number of assayed animals: WT ($n = 298$), *odr-3*(*lf*) ($n = 264$), D175V ($n = 224$), D175 rescue ($n = 157$). Means ± SD are indicated by shaded and vertical bars, respectively. ** and ***Different from wild type at $P < 0.01$ and 0.001, respectively (Fisher's exact test). ###Different between bracketed genotypes (Fisher's exact test).

(Fig. 6b and c), suggesting that *D175V* mutation interferes with ODR-3 trafficking mechanisms conserved across cell types. Next, we examined whether ODR-3 localization to the ASH cilium is required for its function in mediating avoidance of high-osmolarity solutions. To this end, we placed wild-type, *odr-3*(*lf*), or *D175V* homozygous animals inside a ring of 8 M glycerol and counted the number of animals that escaped the glycerol ring within 2 min (Cornils et al. 2016; Fig. 6d). On average, 80% of wild-type animals avoided glycerol and stayed inside the ring (Fig. 6e). In contrast, *odr-3*(*lf*) mutants escaped the ring, and only 40% of *odr-3*(*lf*) animals remained inside the ring after 2 min (Fig. 6e). Similarly to *odr-3*(*lf*) animals, 42% of *D175V* mutants remained inside the glycerol ring after 2 min (Fig. 6e), indicating that *D175V* variant impairs ASH sensory function. Importantly, overexpression of wild-type *odr-3* cDNA in ASH neurons of *D175V* animals partially but significantly rescued glycerol avoidance (Fig. 6e),

confirming that the observed sensory deficit is not due to a second-site mutation.

## D173V, K270R, and A326P patient variants impair ciliary localization of Gαi1 protein in human cells

We next explored whether the variants with the strongest impact on ODR-3 localization demonstrated similar localization defects in human cells. Gαi1 was previously reported to localize to primary cilia of cultured mouse fibroblasts (Singh et al. 2015). We first examined subcellular localization of wild-type human Gαi1 by transfecting HEK293T cells with *eGFP*-tagged *GNAI1* cDNA. In line with the published data, human Gαi1^WT::eGFP was present inside the cilia as well as at the plasma membrane and Golgi (Gabay et al. 2011; Fig. 7a and b). Strikingly, *D173V*, *K270R*, and *A326P* patient variants orthologous to the examined ODR-3 mutants significantly reduced intraciliary levels of the encoded proteins relative

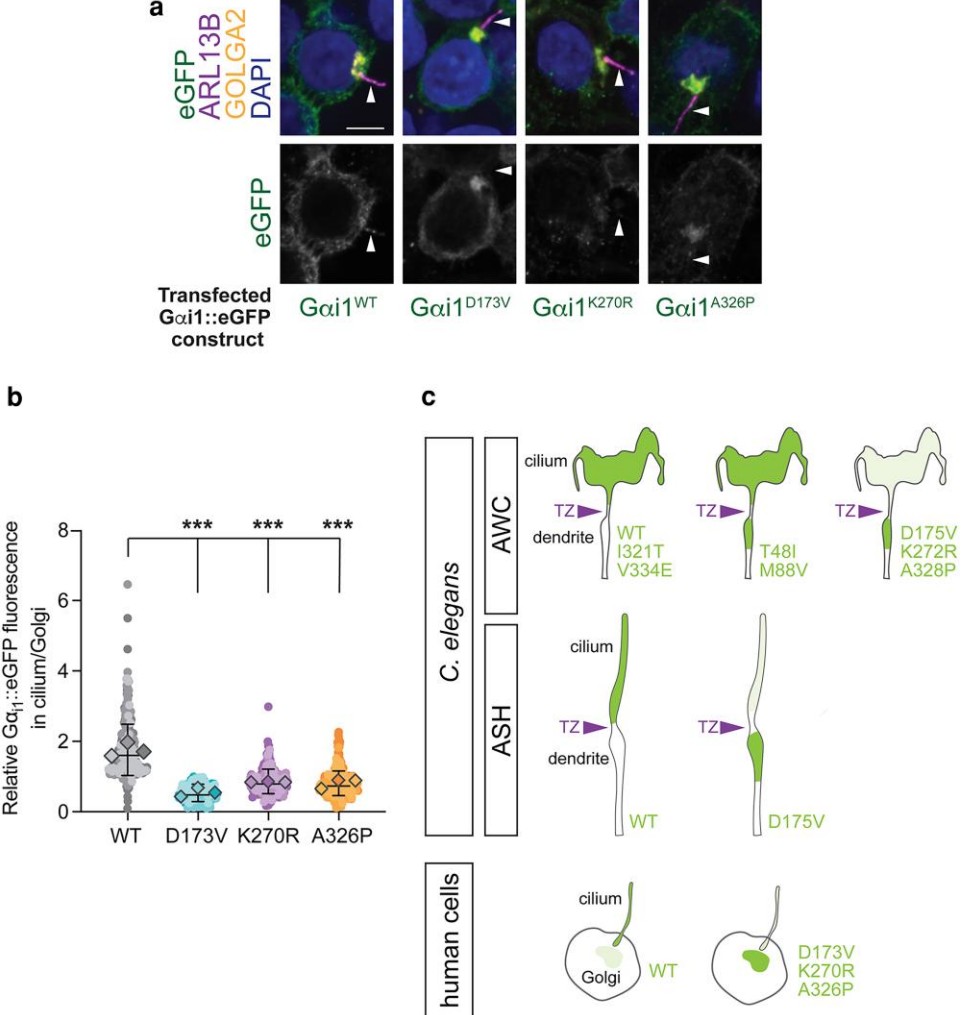

**Fig. 7.** The *D173V*, *K270R*, and *A326P* variants in *GNAI1* impair cilia localization of the encoded proteins in human cells. a) Immunofluorescence images of fixed HEK293T cells transfected with the indicated eGFP-tagged Gαi1 variants. Arrowheads mark cilia labeled with anti-ARL13B antibody. DAPI and anti-GOLGA2 label DNA and Golgi, respectively. Scale: 5 μm. b) Quantification of relative eGFP fluorescence in cilium vs Golgi for the indicated eGFP-tagged Gαi1 variants. Individual biological replicates for each variant are shown in different shades of the corresponding color. Diamonds: means of individual biological replicates. Total mean ± SD across all biological replicates is indicated by horizontal and vertical bars, respectively. Total number of cells analyzed for each Gαi1 variant: WT ($n = 322$), D173V ($n = 220$), K270R ($n = 260$), A326P ($n = 315$). \*\*\*Different from Gαi1$^{WT}$ at $P < 0.001$ (Kruskal–Wallis with Dunn's multiple comparisons test). c) Graphical summary of localization patterns for the indicated ODR-3 and Gαi1 variants in *C. elegans* (AWC and ASH neurons) and mammalian cells, respectively. TZ, transition zone.

to Golgi (Fig. 7a to c), demonstrating that the impact of these variants on Gα protein trafficking is evolutionarily conserved.

## Discussion

### Orthologous *GNAI1*-disorder patient variants differentially impact *odr-3* function

Here, we demonstrate that *GNAI1*—the causal gene in a recently identified NDD—regulates ciliogenesis in 2 human cell lines, suggesting that cilia dysfunction may play a role in the pathogenesis of *GNAI1* disorder. We used *C. elegans* as a whole-animal model to functionally classify a subset of orthologous *GNAI1*-disorder mutations in vivo and found variant- and cell-specific effects on the examined phenotypes (Fig. 7c; Table 1). Overall, orthologous missense mutations that altered conserved amino-acid residues in the nucleotide-binding interface of Gα (ODR-3: *T48I*, *K272R*, and *A328P*) caused defects in both examined ODR-3-dependent phenotypes—AWC cilia morphology and chemotaxis behavior. Similarly, the *V334E* variant, which maps to the terminal α5 helix of Gα ODR-3, caused a mild

reduction in AWC cilia size and significantly impaired attraction to benzaldehyde. The α5 helix of Gα proteins participates in GPCR binding and receptor-catalyzed nucleotide exchange (Marin et al. 2002; Oldham et al. 2006; Alexander et al. 2014; Masuho et al. 2023). Furthermore, the *V332A* substitution was shown to significantly destabilize the GDP-bound state of Gαi1 (Sun et al. 2015). Therefore, additional in vitro studies could shed more light on GPCR binding efficiency and nucleotide exchange rates of V332E Gαi1 variant present in *GNAI1*-disorder patients.

On the other hand, variants altering conserved amino acids in domains of unknown function (ODR-3: *M88V* and *I321T*) had no impact on either AWC cilia morphology or behavior. Recent studies have begun to highlight challenges associated with interpreting the pathogenicity of de novo variants across developmental disorders (Mani 2017). For example, 1 study estimated that only ~13% of de novo missense variants contribute to the risk of autism-spectrum disorder (Iossifov et al. 2014). Therefore, functional studies in additional in vivo models and cellular contexts, in which Gαi1/ODR-3 functions, are needed to confirm the

potentially benign or disease-contributing nature of *M88V* and *I321T* variants.

Intriguingly, the *D175V* variant, which caused marked mislocalization of the encoded ODR-3 protein from the cilium to the distal dendrite, was associated with significantly reduced AWC cilia area, yet had no impact on AWC-mediated attraction to benzaldehyde. In stark contrast, the same *D175V* variant did not affect cilia length in ASH neurons but compromised ASH-mediated glycerol avoidance, demonstrating that the functional impact of the same variant varies depending on the cellular context. Context-dependent differences in functional effects of patient variants have also been reported for other disorders (Feng et al. 2017; Wong et al. 2019; Muntean et al. 2021; Wang et al. 2022; Di Rocco et al. 2023), highlighting the importance of modeling patient mutations in multiple experimental systems and disease-relevant contexts to gain a comprehensive understanding of the underlying disease mechanisms. *GNAI1*-disorder patients exhibit symptoms that range in severity; however, there is no clear correlation between the *GNAI1* variants and the associated phenotypic features at this time (Muir et al. 2021). Cell-specific effects of certain de novo variants (eg *GNAI1*: *D173V*) and timing of their appearance may at least in part contribute to disease burden in patients (Acuna-Hidalgo et al. 2016).

## Modeling orthologous *GNAI1*-disorder missense variants in *C. elegans* provides additional insight into cell-specific functions of ODR-3.

Three of the examined orthologous variants (ODR-3: *D175V*, *K272R*, and *A328P*) caused qualitatively and quantitatively similar defects in the localization of the encoded ODR-3 proteins. Specifically, mutant ODR-3 variants accumulated in the distal dendrite and were present at markedly reduced levels inside the AWC cilium, where wild-type ODR-3 normally resides. Both *K272R* and *A328P* variants alter amino acids that come into direct contact with the nucleotide and, therefore, likely disrupt these interactions and impair Gα signaling function in addition to altering localization of the encoded proteins. Consistently, prior in vitro work showed that a different missense mutation at the A326 position of Gαi1 (*A326S*) accelerated dissociation of GDP from the αi1β1γ1 heterotrimer by >200-fold (Posner et al. 1998). Although found in the immediate vicinity of the nucleotide-binding motif (see Fig. 2b; Kant et al. 2016), the *D175* residue does not directly bind GDP. The finding that the *D175V* mutant can still respond to benzaldehyde implies that the *D175V* variant disrupts ciliary trafficking of Gα but not its ability to signal and puts forward a hypothesis that confirmation of the nucleotide-binding pocket may be important for efficient translocation of ODR-3/Gαi1 into the cilium. Improved targeting of ODR-3[A328P] protein to the cilia in *mks-5(lf)* mutants that have compromised TZ is consistent with this hypothesis and suggests that the *A328P* mutation impedes the translocation of ODR-3 across the TZ barrier. Based on the results from the BiFC assay, ODR-3[A328P] appears to still interact with UNC-119—a key mediator of Gα ciliary transport, although possibly to a lesser degree than ODR-3[WT]. Since the BiFC signal is binary (either on or off), in the future, it would be important to directly examine the stability of UNC-119 complexes with ODR-3/Gαi1 proteins carrying *A328P/A326P* as well as *D175V/D173V* and *K272R/K270R* patient substitutions using biochemical approaches to determine whether the same mechanisms underlie trafficking defects of all 3 variants.

We also found that although the *D175V* variant caused mislocalization of the encoded ODR-3 protein from the cilium to the dendrite, it did not alter chemotaxis toward benzaldehyde. This observation led us to conclude that ODR-3 localization to the cilium is not necessary for its function in mediating primary sensory responses to volatile odorants detected by AWC. Similarly, a recent study demonstrated that diacetyl receptor ODR-10 and Gα ODR-3, which also functions downstream from ODR-10 to mediate attraction toward diacetyl, translocate from the cilium to dendritic branches in AWA olfactory neurons upon disruption of intraflagellar transport (IFT) (Philbrook et al. 2024). Interestingly, IFT mutants retain normal primary responses to diacetyl but exhibit defects in habituation and desensitization (Larsch et al. 2015; Philbrook et al. 2024). It will be of interest to determine whether sequestration of ODR-3[D175V] in the distal dendrite similarly impacts desensitization and habituation responses of AWA and AWC neurons or primary sensory responses to volatile odorants other than benzaldehyde.

Finally, in contrast to AWC, ODR-3 localization to the ASH cilium is required for its function in glycerol avoidance, as *D175V* homozygous mutants exhibit marked deficits in this ASH-mediated sensory behavior despite having normal cilia length. Recent electron microscopy studies in the human and mouse cortex uncovered the remarkable complexity of cilia interactions with the cellular environment in the brain (Ott et al. 2024; Wu et al. 2024). Neuronal cilia were shown to be immersed in dense neuropil and adjacent to axonal and dendritic processes at several points along their length (Ott et al. 2024). Occasionally, tips of neuronal cilia were also found to be enveloped by astrocytes. While the exact nature and functional significance of these interactions remain a mystery, synapses were frequently observed adjacent to cilia in mouse and human cortical tissue (Ott et al. 2024; Wu et al. 2024), and 1 study even reported axo-ciliary synapses between brainstem axons and cilia of hippocampal CA1 neurons (Sheu et al. 2022). Collectively, these observations propose an exciting hypothesis that primary cilia may be a major player in neural circuit modulation. Therefore, mislocalization of Gαi1—a key transducer of GPCR signaling that modulates many aspects of neuronal physiology—from cilia to ectopic subcellular locations may have a sizeable impact on local neural connectivity as well as neuron-neuron and/or neuron-glia communication. It would be worthwhile to determine whether the patient Gαi1 variants that exhibit reduced ciliary levels in *C. elegans* neurons and human epithelial cells are similarly excluded from cilia of distinct neuron classes in the mammalian brain and to classify their impact on neuronal cilia length and circuit function.

In conclusion, our results highlight the power of *C. elegans* as a whole-organism system to rapidly determine functional impact of missense patient variants in conserved Gα residues and thus prioritize functionally consequential mutations for further mechanistic studies in mammalian models and/or additional disease-relevant contexts. Moreover, orthologous disease-associated mutations in model organisms like *C. elegans* can provide new insights into the normal function of the impacted genes.

## Data availability

Strains and plasmids used in this work are listed in Supplementary Tables 1 and 2, respectively, and are available upon request. All data necessary for confirming the conclusions of the article are present within the article, figures, table, and Supplementary material.

Supplemental material available at GENETICS online.

## Acknowledgments

We are grateful to members of the Nechipurenko lab for critical comments on the manuscript and to Ryan Breitenbach for

technical assistance and maintenance of strains. Some strains were provided by the CGC, which is funded by NIH Office of Research Infrastructure Programs (P40 OD010440).

## Funding

This work was supported by the Charles H. Hood Foundation (Child Health Research Award to I.N.) and in part by the National Institutes of Health (R35 GM155316 to I.N.).

## Conflicts of interest

The authors declare no conflicts of interest.

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

*Editor: X. Tong*
