## [Peer Review File · Genetics]

Functional classification of GNAI1 disorder variants in *C. elegans* uncovers conserved and cell-specific mechanisms of dysfunction

Inna Nechipurenko, Rehab Salama, Eric Peet, Thomas Morrione, Sarah Durant, Maxwell Seager, Madison Rennie, and Suzanne Scarlata

NOTE: The reviews and decision letters are unedited and appear as submitted by the reviewers.

In extremely rare instances and as determined by a Senior Editor or the EIC, portions of a review may be redacted. If a review is signed, the reviewer has agreed to no longer remain anonymous.

The review history appears in chronological order.

Review Timeline:

Submission Date:	2025-08-17
Editorial Decision:	2025-09-15
Revision Received:	2025-09-24
Accepted:	2025-09-26

September 15, 2025
RE: GENETICS-2025-308494

Dear Dr. Nechipurenko:

I am pleased to accept your manuscript titled "Functional classification of GNAI1 disorder variants in *C. elegans* uncovers conserved and cell-specific mechanisms of dysfunction" for publication in GENETICS, pending minor revision.

Please submit your revision along with a brief description of how you modified the manuscript in response to the reviewers' concerns and suggestions (which can be viewed at the bottom of this email. Most important are:

1. Add a concise description of AWC ciliary anatomy and a graphical model or illustration summarizing the localization patterns of ODR-3 variants.
2. Discuss how different ODR-3 variants show distinct effects in different chemosensory neurons (e.g., ASH vs. AWC). Consider whether these findings provide insight into the diverse phenotypes observed in patients with GNA1 missense variants.
3. Provide a clearer explanation of the observed decrease in RIC-8 and UNC-119 binding.
4. Clarify the rationale for choosing specific variants for RIC-8 and UNC-119 binding assays, and for ASH neuron-mediated function.

I expect you should be able to submit a revised manuscript within 30 days. A suitably revised manuscript will be acceptable for publication.

Please ensure that you have included a Data Availability Statement at the end of the Materials and Methods section. Details available at <https://academic.oup.com/genetics/content/prepare-manuscript>. The DAS should include the accession numbers or DOIs of any data you have placed in public repositories, describe supplemental material, include applicable IRB numbers, and may include specifications for how to properly acknowledge or cite the data.

When revising the ms., please make an effort to shorten it, because that almost always improves a manuscript. We urge authors to heed the advice of Strunk and White: "omit needless words"¹. Follow this link to submit the revised manuscript: Link Not Available

Thank you for submitting this story to Genetics.

Sincerely,

Xiajing Tong
Associate Editor
GENETICS

Approved by:
Oliver Hobert
Senior Editor
GENETICS

Reviewer comments:
Reviewer #1 :

The manuscript by Salama et al. validate *C. elegans* as a disease model to study the consequence of variants in GNAI1, a gene associated with human disease. Using transgenic reporters and CRISPR-Cas9 technology, the authors found that specific GNAI1 variants reduce cilia size, affect ODR-3 (*C. elegans* orthologue) cellular localization, and alter AWC and ASH function. The experiments appropriately support the conclusions, and results are clearly presented. The manuscript is well written and organised. However, a few points require attention.

Major points:

1. Experiments line 372 - Is the transition zone required for the ciliary exclusion of other ODR-3 variants or solely A328P? It would be interesting to do this analysis for variants with similar localization to A328P.
2. Supplementary Figure 6a&b - There is quite a noticeable reduction in the binding of RIC-8::VC/ODR-3WT in comparison to RIC-8::VC/ODR-3A328P. However, this is not clearly described and explained. It is not clear why in supplementary figure 7a&b (line 437-459) more attention is given to the BiFC results (intensity) in between UNC-119 and ODR-3A328P, when these are similar to the results presented in supplementary Figure 6a&b. The results show that interaction of RIC-8 and ODR-3 was not abolished. Can a reduction in interaction affect cilia morphology?
3. Page 19 - It is not clear why other variants such ODR-3D175V and ODR-3K272R that show cilia localization defects were not tested to confirm whether the UNC-119 and ODR-3 complex was also affected. This could give more information about the mechanism involved in the different variants.
4. Page 21 - It is not clear why were the other variants not tested for ASH defects?
5. In future manuscripts, it would be interesting to explore why ODR-3D175V presents cilium localization in defects in AWC and ASH but only present chemotaxis defects in ASH neurons.

Minor:

Figure 1 - why is there a n number (n=518 RPE-1, n=4600 HEK293T) so different in between the two cell lines?

Line 396 - The authors mentioned "...exhibited marked reduction in AWC cilia size compared to wild type". I suggest rephrasing to cilia area as in Fig 5b since size can be understood as length and in some images in Fig 5a does not appear to be the case.

Fig 5a - Purple arrows are white.

Figure 1d - The authors suggest the impact of GNAI1 KD on ciliation of HEK293T cells appears to be qualitatively milder relative to RPE-1 cells (approx. 30% vs 40%), however, the number of ciliated cells in siControl is also relatively low in comparison to RPE-1 cells (approx. 40% vs 80%). Provide explanation and indicate whether this is a limitation of the methodology

Figure 3a, 4a - A schematic of the cilia base and dendrite in the images provided would be helpful.

Supplementary Figure 1 - a), b) the controls are the same used in Fig.1c and Fig.1e, respectively. The cilia and expression analyses were done at the same time for GNAI1#1 and GNAI1#2? Clarify whether the experiments were done simultaneously. If so, it would be more accurate to represent, at least the expression data, in the same graph/figure instead of having the GNAI1#2 data in supplementary data.

Figure 1 and Supplementary Figure 1- In the figure legend, it mentions "Summary data represent > 3 replicates". Specify the number of replicates analysed for siControl and siGNAI.

Chemotaxis assay - It would be important to test other chemicals detected by ODR-3 (AWC) in the different variants in a future manuscript. Maybe there is no phenotype regarding benzaldehyde, but the variants may affect the detection of other molecules.

Reviewer #2 :

Salama, Nechipurenko and colleagues develop a powerful strategy to study variants of uncertain significance for the GNA1 disorder using mammalian cell culture and a *C. elegans* in vivo model. Using this strategy, Salama et al show that GNA1 knockdown in HEK293T and RPE-1 cells reduces the number of ciliated cells, suggesting that GNA1 disorder may be a ciliopathy. In *C. elegans*, only one (I321T) engineered orthologous mutation is harmless while others have different harmful effects on protein subcellular localization, ciliary morphology, and chemotaxis behavior. Three variants with the strongest functional impact on ODR-3 were engineered into human Galphai1 and expressed in HEK293T cells - D173V, K270R, and A326R patient variants all affect ciliary localization, demonstrating evolutionary conservation and the power of authors' dual approach. The data and its presentation are strong and convincing. The scientific rigor is high. For example, to assess impact of orthologous missense VUS on protein subcellular localization, authors employ complementary approaches using endogenously tagged ODR-3 and transgene overexpression. The manuscript is well-written, easy to follow, will appeal to those interested in modeling human genetic disease, neurodevelopmental disorders, cilia biology, and behavior, making this work a great fit for the broad readership of Genetics.

Specific comments:

Ciliary localization section: For non-cilia reader, it would help to explain AWC ciliary anatomy: wing, transition zone, periciliary membrane compartment, distal dendrite in one place, to provide context for results.

P15: Split-wrmScarlet strategy: for a non-*C. elegans* aficionado, a brief explanation is needed

None of the mutations completely abolished ciliary/PCMC localization or disrupted earlier steps of protein localization. A328P appears to have the most profound impact on ODR-3 localization and most closely resembles the *odr-3* null mutation, yet does

not affect interactions with RIC-8 or UNC-119. Any published clinical descriptions of patients with GNAI1 missense variants that could be explained by differences in phenotype in *C. elegans*? This may be an interesting discussion point.

Reviewer #3 :

Mutations in GNAI1 have been recently linked to a neurodevelopmental disorder. However, the mechanisms by which these patient mutations disrupt neuronal development or function remain unknown. In this manuscript, Salama et al. expertly use *C. elegans* as a live model organism to conduct a functional analysis of seven patient-derived GNAI1 variants. They provide compelling evidence that most of these mutations affect the ciliary localization of ODR-3 (the ortholog of GNAI1 in *C. elegans*) and cilia related sensory function, thereby linking GNAI1 mutations to ciliogenesis defects and offering a plausible pathogenic mechanism for the observed neurological symptoms.

I have the following questions and suggestions:

1. Figure 3 is a key figure that details the subcellular localization defects of the ODR-3 variants. To make these important results easier for the reader to interpret, I suggest adding a graphical model or illustration summarizing the localization pattern (e.g., cilium, dendrite) for each variant type.
2. The neuron-specific effect of the D175V variant is intriguing. Do the authors have any hypotheses for why it specifically impairs ASH neuron-mediated response but not AWC neuron function? Could this specificity be related to differences in GPCRs, effectors, or other signaling partners between these neuronal types?
3. The M88V and I321T variants show no defects in AWC neuron-mediated chemotaxis. Could they potentially exert a function in other sensory neurons? Similar to the approach taken with D175V, it might be informative to test their effect on ASH neuron-mediated sensory responses to determine if their impact is also neuron-specific.
4. Role of the Transition Zone: The data show that the transition zone functions as a barrier for the ciliary entry of the ODR-3 A328P variant. Since the D175V and K272R variants also accumulate in the dendrite, is their ciliary entry also affected by the transition zone? Furthermore, given that MKS and NPHP represent two major molecular modules at the transition zone, would loss of NPHP components also affect the ciliary entry of these variants?

Response to Reviewers

We would like to thank the reviewers for their thoughtful and insightful comments that helped us improve the quality and clarity of the manuscript. Our responses to editors' and reviewers' concerns and suggestions are summarized below.

Editors' comments

1. *Add a concise description of AWC ciliary anatomy and a graphical model or illustration summarizing the localization patterns of ODR-3 variants.*

We have added a diagram of the AWC neuron and a magnified view of the AWC distal dendrite including the cilium, TZ, and PCMC **to Fig. 3 (new panel a)**. We also added illustrations summarizing localization of ODR-3 variants in AWC neurons to **Fig. 3b and Fig. 4a and b**. Finally, we included a graphical summary showing localization of ODR-3 and G α i1 variants to **Fig. 7 (new panel c)**.

2. *Discuss how different ODR-3 variants show distinct effects in different chemosensory neurons (e.g., ASH vs. AWC). Consider whether these findings provide insight into the diverse phenotypes observed in patients with GNA1 missense variants.*

We have clarified the rationale for choosing specifically the D175V variant for comparative analysis in AWC and ASH neurons and expanded discussion as to why this variant exerts distinct effects on AWC and ASH neuron function (**lines 518-523**). We also added discussion about how cell-specific impacts of patient mutations could potentially contribute to diversity of phenotypes observed in patients (**lines 587-591**).

3. *Provide a clearer explanation of the observed decrease in RIC-8 and UNC-119 binding.*

We clarified the explanation of the BiFC results and included sections that directly discuss a decrease in percentage of animals that exhibit BiFC when ODR-3^{A328P} variant is co-expressed with RIC-8 (**lines 429-437**) or UNC-119 (**lines 466-474**). We also added statistics to **Supplementary Figures 6b and 7b** in further support of these conclusions.

4. *Clarify the rationale for choosing specific variants for RIC-8 and UNC-119 binding assays, and for ASH neuron-mediated function.*

We clarified the rationale for choosing specific variants for BiFC assays (**lines 424-427; lines 454-456**) and ASH behavioral assay (**lines 518-523**)

Summarized below are responses to reviewers' questions and concerns that are not already covered in responses to the editors.

Reviewer 1:

Major points:

1. *Is the transition zone required for the ciliary exclusion of other ODR-3 variants or solely A328P? It would be interesting to do this analysis for variants with similar localization to A328P*

We agree that it would be interesting to examine whether cilia entry of other variants with similar localization defects would be improved upon TZ disruption and would, therefore, suggest that similar mechanisms underlie localization defects of all variants (**line 377**; **lines 604-616**).

For major points 2-5, please see responses to the editors above.

Minor points:

- *Figure 1 - why is there a n number (n=518 RPE-1, n=4600 HEK293T) so different in between the two cell lines?*

HEK293T cells are more confluent than RPE-1 cells and grow in multi-layer patches; therefore, the total number of cells per each quantified image is greater for HEK293T than RPE-1 cells. Thus, the total number of quantified cells is also greater.

- *The authors mentioned "...exhibited marked reduction in AWC cilia size compared to wild type". I suggest rephrasing to cilia area as in Fig 5b.*

Done

- *Fig 5a - Purple arrows are white*

We changed the shade of the arrowheads to make purple more obvious to the reader.

- *Figure 1d - The authors suggest the impact of GNAI1 KD on ciliation of HEK293T cells appears to be qualitatively milder relative to RPE-1 cells (approx. 30% vs 40%), however, the number of ciliated cells in siControl is also relatively low in comparison to RPE-1 cells (approx. 40% vs 80%). Provide explanation and indicate whether this is a limitation of the methodology.*

We included explanation (**lines 312-315**).

- *Figure 3a, 4a - A schematic of the cilia base and dendrite in the images provided would be helpful.*

Done (please see response to the editors)

- *Supplementary Figure 1 - a), b) the controls are the same used in Fig. 1c and Fig. 1e, respectively. The cilia and expression analyses were done at the same time for GNAI1#1 and GNAI1#2? If so, it would be more accurate to represent, at least the expression data, in the same graph/figure instead of having the GNAI1#2 data in supplementary data.*

Thank you for this suggestion. We combined KD and qPCR data for siGNAI2 with data in Figure 1, as the experiments were all done in parallel.

- *Figure 1 and Supplementary Figure 1. Specify the number of replicates analysed for siControl and siGNAI.*

Done

- *Chemotaxis assay - It would be important to test other chemicals detected by ODR-3 (AWC) in the different variants in a future manuscript. Maybe there is no phenotype regarding benzaldehyde, but the variants may affect the detection of other molecules.*

We have added this point to Discussion (**line 626-628**).

Reviewer 2:

All concerns and suggestions brought up by reviewer 2 were addressed in responses to the editors above.

Reviewer 3:

For points 1 and 2, please see response to the editors above.

3. The M88V and I321T variants show no defects in AWC neuron-mediated chemotaxis. Could they potentially exert a function in other sensory neurons? Similar to the approach taken with D175V, it might be informative to test their effect on ASH neuron-mediated sensory responses to determine if their impact is also neuron-specific.

Please see response to the editors above for the rationale behind choosing specifically *D175V* for comparative analysis in AWC and ASH neurons and reasons as to why this variant disrupts ASH but not AWC primary responses. Since *M88V* and *I321T* variants do not disrupt localization or signaling function of the encoded proteins (i.e. *M88V* and *I321T* mutants respond to benzaldehyde), we do not expect them to have an impact on primary ASH sensory responses that similarly require intact ODR-3 function. However, we appreciate the reviewer's point, as these variants could potentially disrupt interactions of ODR-3 with partner proteins in other cellular context(s) and thereby impact ODR-3 function there (**lines 575-577**).

4. The data show that the transition zone functions as a barrier for the ciliary entry of the ODR-3 A328P variant. Since the D175V and K272R variants also accumulate in the dendrite, is their ciliary entry also affected by the transition zone? Furthermore, given that MKS and NPHP represent two major molecular modules at the transition zone, would loss of NPHP components also affect the ciliary entry of these variants?

Please see response to Reviewer 1 (point 1). Since mutations in individual components of the MKS (except for *mks-5*) or NPHP module in *C. elegans* are typically not sufficient to compromise TZ integrity (Jensen et al., 2015), we chose *mks-5(tm3100)* mutant for this analysis, because it is a core component of the TZ and has been shown to severely compromise its integrity (**lines 382-383**). Unless the NPHP module is directly involved in regulating ciliary entry of ODR-3, we would not expect to see changes in A328P localization in single mutants for NPHP components.

September 26, 2025
RE: GENETICS-2025-308494R1

Dr. Inna Nechipurenko
Worcester Polytechnic Institute
Biology and Biotechnology
60 Prescott Street
Worcester, Massachusetts 01605

Dear Dr. Nechipurenko:

Congratulations, your manuscript titled "Functional classification of GNAI1 disorder variants in *C. elegans* uncovers conserved and cell-specific mechanisms of dysfunction" is accepted for publication in GENETICS! Many thanks for submitting your research to the journal.

As part of our efforts to make titles of articles published in GENETICS more accessible to our broad readership, we often suggest different titles for accepted manuscripts. We offer these variants for your consideration though we'll use whatever title you include in the final version of your manuscript that you submit to the Editorial Office.

Title suggestions:

Ciliogenesis in *C. elegans* and humans requires the G-protein G i1

Human GNAI1 disorder: a ciliopathy caused by a defective G-protein conserved in *C. elegans*

Human GNAI1 disorder: a ciliopathy revealed by functional analysis of disease variants in *C. elegans*

To Proceed to Publication:

1. Format your article according to GENETICS style: <https://academic.oup.com/genetics/pages/author-guidelines>
2. Ensure that you comply with data and community resource citation guidelines: <https://academic.oup.com/genetics/pages/author-guidelines#section-5-9-2>
3. Upload your final files at <https://genetics.msubmit.net>
4. Add oupsupport@scipris.com and genetics.oup@novatechset.com (or the domains @scipris.com and @novatechset.com) to your email program's "safe senders" list. You will be contacted by both at various points during the production process.

Notes:

- Your currently-accepted manuscript (unedited, as submitted, reviewed, and accepted) will be published at GENETICS and deposited into PubMed as an Advance Access article. Notify sourcefiles@thegsajournals.org before signing your license if you do not wish to publish your article via Advance Access.
- We invite you to submit an original color figure related to your paper for consideration as cover art. Please email your submission to the editorial office or upload it with your final files. You can submit a small-sized image for evaluation, and if selected, the final image must be a TIFF file 2513px wide by 3263px high (8.375 by 10.875 inches; resolution of 600ppi). Please avoid graphs and small type.
- After files are sent to Oxford University Press we use SciPris to manage article licensing and payment. If you do not have a SciPris account, you will receive an email from no-reply@scipris.com to sign up to use Oxford University Press' author portal. After logging in, follow the online instructions to sign your license and arrange any payment due.

If you have any questions or encounter any problems while uploading your accepted manuscript files, please email the editorial office at sourcefiles@thegsajournals.org.

Sincerely,

Xiajing Tong

Associate Editor
GENETICS

Approved by:
Oliver Hobert
Senior Editor
GENETICS